# Biodegradable Polyesters and Low Molecular Weight Polyethylene in Soil: Interrelations of Material Properties, Soil Organic Matter Substances, and Microbial Community

**DOI:** 10.3390/ijms232415976

**Published:** 2022-12-15

**Authors:** Jana Šerá, Florence Huynh, Faith Ly, Štěpán Vinter, Markéta Kadlečková, Vendula Krátká, Daniela Máčalová, Marek Koutný, Christopher Wallis

**Affiliations:** 1Department of Environmental Protection Engineering, Faculty of Technology, Tomas Bata University in Zlín, Nad Ovčírnou 3685, 760 01 Zlín, Czech Republic; 2Polymateria Limited, Imperial College I-HUB, White City Campus, 84 Wood Lane, London W12 OBZ, UK; 3Centre of Polymer Systems, University Institute, Tomas Bata University in Zlín, Tr. T. Bati 5678, 760 01 Zlín, Czech Republic

**Keywords:** biodegradation, polyesters, polyethylene, soil organic matter, fungi, bacteria, SEM

## Abstract

Conventional and also biodegradable polymer microplastics have started to be broadly present in the environment, if they end up in soil, they may influence both abiotic and biotic soil properties. In this study, the interactions of polyethylene wax together with three biodegradable polyesters PLA, PHB and PBAT with a soil matrix were investigated over a 1-year incubation period. Soil organic matter content was measured using UV–VIS, the microbial biomass amount was measured using qPCR, the mineralisation of polymers was measured using UGA 3000, the surface of polymers was observed with SEM, live/dead microorganisms were determined by fluorescent microscopy and microbial consortia diversity was analyzed using NGS. The amount of humic substances was generally higher in incubations with slowly degrading polyesters, but the effect was temporary. The microbial biomass grew during the incubations; the addition of PHB enhanced fungal biomass whereas PE wax enhanced bacterial biomass. Fungal microbial consortia diversity was altered in incubations with PHB and PBAT. Interestingly, these two polyesters were also covered in biofilm, probably fungal. No such trend was observed in a metagenomic analysis of bacteria, although, bacterial biofilm was probably formed on the PE520 surface. Different methods confirmed the effect of certain polymers on the soil environment.

## 1. Introduction

Microplastics have become a major concern of late. They enter the environment, interact with both the geosphere and the biosphere and become a part of biogeochemical cycles just like other substances [1,2,3]. Soil represents a medium where microplastics can accumulate [4,5,6,7]. They can originate directly from the materials used intentionally in agriculture e.g., mulching films, coated fertilizers and pesticides, or they can be unintentionally introduced along with materials used to increase fertility e.g., sediments and wastewater treatment sludge [8,9,10]. Regarding their interaction with soil microorganisms and consequently their tendency to accumulate, microplastics can be roughly divided into conventional (PE, PP, …) and biodegradable (PHA, PBAT, …). Conventional polymers are non-biodegradable, although partial degradation in soil could be achieved. Biodegradation is mostly limited to shorter and flexible chains in the amorphous region of the polymer backbone [11,12,13]. Biodegradable polymers are most often polyesters, mostly aliphatic, rarely with a small portion of aromatic constituents [14,15,16,17]. The ester bonds can be cleaved by microorganisms and the material is degraded in soil, typically over several months [18,19,20]. The majority of carbon released from polymer chains during biodegradation is transformed into CO_2_, but some of the carbon can be incorporated into biomass or enter the soil organic carbon pool [21].

The ability of microplastics to influence soil microbial population has been investigated in several studies. Although microplastics did not generally seem to cause any adverse effect on microorganisms, their addition to soil changed the structure of the microbial community, the overall microbial biomass, respiration, and the activity of some enzymes etc [22,23,24,25]. According to several studies, it seems that biodegradable microplastics provoke different changes compared to conventional microplastics [26,27], but there are also studies that disagree with these findings [6]. The possible ecotoxicity of polyethylene in relatively high dosages to some microorganisms has been reported recently [28]. The effect on soil microorganisms including both bacteria and fungi in terms of biomass and diversity and the comparison between biodegradable materials and conventional polymer microplastics has not been studied yet.

Microplastics in soil may change soil properties such as water-holding capacity, soil infiltration and water absorption [6,29,30]. These abiotic soil properties together with the shift in microbial consortia composition and biomass may lead to changes in carbon cycling and storage in soil over time [31] and potentially have long term effects on soil organic matter dynamics. There are several studies that deal with the direct effect of microplastics on soil organic matter substances, but most of them concentrate on interactions with dissolved organic matter [32,33,34]. Because microplastics affect microbial activity in soil, their addition may influence soil organic matter turnover efficiency, especially the labile organic carbon pool, which has a relatively fast turnover rate [35,36]. It seems that the priming effect of conventional microplastics is limited [37], but biodegradable polymers may have a considerably greater impact [38].

After its mineralisation, carbon from polymers may contribute to growing CO_2_ levels in the atmosphere. The possibility of its incorporation into microbial biomass and organic matter substances may enhance the carbon neutrality of biodegradable polymers. Both non-biodegradable and biodegradable microplastics can influence the microbial community composition, soil structure, and fractions of organic carbon in soil, but some fundamental differences can be expected between the two groups of materials. Here, samples of selected non-biodegradable and biodegradable materials and their interactions with soil were studied with the aim to investigate the network of relationships between microplastic transformation, organic soil matter and the microbial community.

## 2. Results

### 2.1. Properties of Studied Materials

A series of biodegradable polyesters together with one PE wax were selected to study their biodegradation and the effect of their material properties in interaction with soil organic matter substances and soil microorganisms. Crystallinity and molecular weight were found in the literature for PLA ad PHB and were measured for the PBAT and PE samples [39]. The properties are summarized in Table 1. he PE sample had a molecular weight close to 10,000 g/mol; such materials should contain fractions of short-chain polymers that could be metabolized by microorganisms [40]. The degree of crystallinity of polyethylene generally determines the resistance of the material, the impact, tensile strength, and the permeability for gasses and stiffness together with resistance to biodegradation [11].

### 2.2. Mineralisation of Materials

Levels of CO_2_ produced during the incubations were measured over 1 year to quantify the mineralisation of carbon in polymers by both biotic and abiotic degradation in the soil environment. The results can be seen in Figure 1.

PHB biodegradation was fast from the very beginning of the incubation, PBAT exhibited a marked lag phase until about 4.5 months, when the mineralisation started to grow rapidly. The metabolism of microorganisms was limited during the preparing lag phase, so mineralisation was decreasing for the first 6 weeks. It can be assumed that the microorganisms started to be more active after this phase. PLA mineralisation was growing quite constantly. Both PBAT and PLA mineralisation reached almost 25%, PHB was almost completely consumed after 1 year. PE representative mineralisation reached almost 15%. Mineralisation was probably affected by its rather low crystallinity and molecular weight. The mineralisation rate of the PE sample was relatively fast at the beginning of the experiment, but then slowed after 100 days; we suggest that this is because a small portion of the short-chain, low-crystalline PE chains was easily metabolised, leaving behind longer chain, high-crystalline polymer parts, which are less accessible to biological attack. In addition, the PE wax sample, unlike the biodegradable polymer samples, had no carbonyl groups, such as carboxylic ester groups, which are well-known to be essential for bacterial and fungal biodegradation.

### 2.3. Microscopic Observation

PHB was almost completely mineralized after 6 months of incubation. The mineralisation curve was steepest in the first weeks of the experiment but a dense biofilm was formed later, between the first month and the sixth month of the incubation based on SEM photographs. Several filaments and cracks were observed after 1 month of incubation, and mineralisation had already reached 30% at the same time. The cracks were formed preferably under filaments. Judging according to their size and morphology, the filaments could be assigned to some fungi species. Since the mineralisation process was already in an advanced stage, it can be assumed that microorganisms in the soil, even if they were not attached to the material surface, degraded the material with the action of their extracellular enzymes. As the abiotic degradation of polyesters under mild temperatures is limited [41], the results seen must be assigned to the action of microorganisms and their enzymes. Fungal filaments and the deterioration of the surface were also observed on the PBAT surface after only 1 month. A net of the filaments and cracks in their vicinity was then formed over 6 months. The microorganisms probably consumed the degradable part of the polymer material before the end of the incubation. No microorganisms were observed in photos taken after 1 year. There were cracks in the material, which probably copied the decomposed filaments. This has also been observed in our previous study [42]. The microorganisms on the surface of these two polyesters were mostly fungi, based on their morphology. No microorganisms were attached to the PLA surface as assumed. PLA biodegradation in soil is very limited, even though the material is regarded as compostable [43,44]. Microbial biofilm was already formed on the surface of PE after 1 month of incubation. The surface of this PE attracted microbial cells, which formed a dense biofilm. The microorganisms were probably from the group *Actinobacteria* based on their size and shape.

The microorganisms were also visible on fluorescent microscopy photos. Microorganisms were stained using the live/dead method and mostly appeared as green, which means that they were actively metabolizing. Microbial biofilm was observed on the surface of PHB, PBAT and PE by fluorescent microscopy, which agreed with SEM observations. Only single colonies were observed on the surface of PLA. The results are visible in Figure 2 and Figure 3.

### 2.4. Biomass of Microorganisms

Total bacterial and fungal biomass in the incubations was investigated to evaluate the possible incorporation of carbon from decomposed polymers into the microbial biomass.

Bacterial biomass was growing constantly in both polymer and blank incubations, but the bacterial counts did not differ between the samples at the beginning of the experiment. In the later phase (6 to 12 months) bacterial biomass was slightly higher in incubations with samples than in blank incubations. The biomass of the PE sample was significantly higher than any other sample after 12 months of incubation but was still in the same range as the other samples. The biomass of bacteria was generally growing during the experiment, except for in the blank sample, where the biomass slightly dropped between 6 and 12 months. A possible explanation for this drop may be that the bacterial biomass in the blank sample proliferated quickly at the beginning of the incubation period and had limited access to sources of energy in the later phase, so the fraction of these cells was decomposed and the bacterial biomass in the blank consequently decreased.

It can be assumed that the polymer-degrading bacteria must possess specific metabolic pathways to degrade the materials. The mass of the degraders was probably not high enough to affect the total number of bacteria in the incubations. The total bacterial biomass did not increase significantly, even in the incubations with samples where microbial cells and filaments were observed on the polymer surface.

Fungi biomass was growing constantly during the whole incubation period but did not differ from the blank for most of the samples except PHB. For some time points and materials, the fungi biomass was even lower than in the blank. The significant growth of fungal biomass was observed only in the PHB incubation. There, fungi biomass was about two times higher than in the blank and other samples after 6 months. The addition of PHB to soil positively affected the growth of fungi in the incubation, serving as an accessible carbon source. This agrees with the experiment by Zhou et al. [23], which proved that the presence of PHBV microplastics increased soil microbial biomass and activity. In our study, we could conclude that carbon in PHB, which was rapidly degraded from the first days of incubation, has a better ability to be transformed into fungi biomass than the other tested polyesters (Figure 4 and Figure 5).

The efficiency of the qPCR reactions was within an acceptable range (90–110%). The deviation was within 1% for all the reactions.

### 2.5. Analysis of Humic Substances

The amount of humic substances was tracked in incubations with samples and in the blank and also in the soil before incubation. Commercial standards were used to prepare the calibration curves. The elemental composition of commercial standards of fulvic (FA) and humic acids (HA) showed that the carbon content was about 49% for fulvic acids and 42% for humic acids, respectively, which is shown in Table 2. These data agreed with other studies [45]. The deviation in elemental composition data was within 2%.

According to Zumstein et al., partial incorporation of polymer carbon into the substances of soil organic matter can be investigated by measuring the amount of fulvic and humic acids [21]. The spectrophotometric determination of Has and Fas was performed to see the evolution of the humic acids during the incubation period and to find out whether carbon from the tested PE wax and biodegradable polyesters was incorporated into the soil organic fraction.

In all instances (Figure 6 and Figure 7), at first, an increase in the content of humic substances could be seen, especially for the slowly degrading polyesters (PLA, PBAT), but after 12 months a slight decrease could be observed. Probably, the substances formed were relatively labile and were metabolized soon after being formed, promoting the biodegradation of other more labile humic substances from the soil organic carbon pool. An important drop in FA can be seen in the case of PHB, which is characteristic of very fast biodegradation in the soil and induced an important increase in fungi biomass. We may speculate that overgrown fungi were able to metabolize some part of FA.

To determine the degree of condensation of HA the ratio of absorbances at 465 and 665 nm (E_4_/E_6_ ratio) was measured after 6 and 12 months of incubation. It was observed that HA did not change structurally during the incubation (Figure 8). The calculated values of the E_4_/E_6_ ratio indicated a high degree of condensation of HA for all samples and for the blank.

### 2.6. Sequestration of Carbon from the Polymer in the Soil-Biomass System

The amount of carbon that was transferred from the sample to the biomass was calculated based on the amount of biomass in each incubation in comparison to the blank at the end of experiment (after 12 months) using qPCR. Carbon content in bacterial and fungal cells was derived from data found in the literature [46,47,48].

The percentage of carbon transferred from original polymeric materials to newly formed bacterial biomass did not exceed 0.1% (*w*/*w*) in all incubations except for sample PE, where 0.3% of the original carbon was transformed from polymer material into the newly formed biomass of the bacteria.

The biodegradation of polyesters was driven mainly by fungal species. The amount of carbon transferred from polymers during their biodegradation to the newly formed fungal biomass did not exceed 0.5% for PLA and PBAT, and about 3.3% of PHB carbon was transferred from the sample to newly grown fungal cells. This number is rather low considering that almost 95% of the original polymer carbon was mineralised as carbon dioxide at the end of experiment and the amount of biomass was doubled compared to the blank.

Based on these results, the amount of carbon that may be transported into microbial biomass was in the range of several percent. Bacterial biomass was affected in the soil incubation with the PE sample, and fungi biomass was affected in the PHB incubation.

Humic and fulvic acids were formed preferably in polyester incubations with samples with limited CO_2_ and biomass production. When compared to the blank samples, no significant amount of new humic and fulvic substances was formed during the incubations. No new HA/FA were formed in incubation with PE.

### 2.7. Interaction of Polymers with the Soil Microbiome

The bacterial community was highly diverse, which is typical for soils. On the class level (Figure 9), no obvious trends and differences between the samples and the blank incubation could be seen. There was a burst in *Actinobacteria* and *Thermoleophilia* presence after the first month, but this feature also appeared in the blank incubation and disappeared after the third month.

Principal component analysis suggested (Figure 10) that indeed after the first month the bacterial community composition shifted and was very similar for all the samples and different from the blank incubation. This could be a result of the development of sedentary cells on the newly available surface of samples and/or consumption of some more accessible molecules present in the sample materials. Then, after 3 months, the samples of biopolymers moved to the left and differentiated themselves from the PE sample. Interestingly, after 12 months, the bacterial community in all incubations found a new equilibrium and grouped in the left upper corner of the scatter plot. The latter finding suggested that after the perturbation the soil bacterial community was not long-term or irreversible.

The differences were more visible in the case of the fungi community where at the family level it could be seen (Figure 11) that distinct new fungi groups appeared in connection with the presence of some materials, most obviously in the case of the fast-degrading biopolymer PHB with *Herpotrichiellaceae*, *Bionectriaceae*, and PBAT with *Ophiocordycipitaceae*, *Herpotrichiellaceae* and *Bionectriaceae*, which belong to the group of saprophytic fungi that prefer woody substrates, making them an obvious candidate for the rather reduced substrate of PHB. Some of the fungi belonging to *Herpotrichiellaceae* were identified as possible PE degraders with lignin-degrading enzyme activity [49] and some were also capable of degrading aromatic compounds [50]. Some of the *Bionectriaceae* family members may be associated with PE, PHB and PUR degradation, probably by hydrolase or/and lignocellulolytic activity [51].

*Mortierellaceae* and *Microascaceae* saprobic fungi are rather common in soil. *Mortierellaceae* and *Bionectriaceae* representatives were able to easily colonise the PE surface in Nowak et al.’s study [52]. On the contrary, the appearance of *Ophiocordycipitaceae* seems rather surprising because this group is known as a parasite of various arthropods.

Principal component analysis again was successful in distinguishing the samples (Figure 12) with relatively fast degrading PHB and PBAT in the upper half and the more stagnant PE and PLA samples in the lower part of the plot together with the blank samples.

The main portion of the carbon which was released from the investigated microplastics was mineralised into CO_2_. A rather small portion of the polymer carbon became a part of the soil organic matter compounds or contributed to the biomass formation. The data from multiple methods such as SEM and molecular biology analysis suggested that the tested PE wax was partially degraded probably due to its content of low MW molecules and low crystallinity, where the biodegradation could be associated mostly with bacteria. The biodegradation of polyesters PHB and PBAT was mostly related to soil fungi. Polyester PLA did not alter the soil microbial community significantly. On the contrary, the biodegradable polyesters PHB and PBAT caused an increase in some fungal taxa abundances in soil, some of them may be the potential degraders of these polyesters. Fungal taxa, whose abundance grew in PBAT and PHB incubations, have been connected to polymer degradation in the literature. We may speculate that different hydrolases are probably important for the degradation of these polymers by fungi. PE wax and PLA both had a small effect on microbial diversity in soil during the incubation period compared to biodegradable PHB and PBAT.

It seems that the formation of a dense biofilm on the surface of PHB and PE contributed to the growth of the overall biomass in these incubations. The close correlation between mineralisation rate, bacterial/fungal biomass growth and biofilm formation was obvious, especially for PHB and PE samples. An interesting finding was that new organic matter substances were formed more in incubations with limited mineralisation, but even in these cases the portion of the carbon transferred from polymers to the soil organic pool was very low and the effect was transient only.

## 3. Materials and Methods

### 3.1. Experimental Design

The relationship between mineralisation, biomass and biofilm formation, soil organic matter and microbial consortia diversity and the composition of conventional (PE) and biodegradable (PHB, PBAT, PLA) microplastics was analyzed using multiple methods.

First the properties of the raw materials (crystallinity—differential scanning calorimetry (DSC), molecular weight—gel permeation chromatography (GPC), and surface structure—scanning electron microscopy (SEM)) were tested. Soil characteristics were provided by LUFA Spreyer. The elemental composition of humic and fulvic acids standards used to create the calibration curve for soil organic matter analysis was measured.

Three incubation sets were prepared:Set 1 for mineralisation measurement;Five replicants for each polymer and five blank replicants (matrix without polymer) were prepared;Set 2 for microscopy and molecular biology analysis;One incubation flask with multiple specimens for each polymer and one blank flask (matrix without polymer) were prepared;Set 3 for humic substances analysis;Three replicants were prepared for each sampling interval and polymer together with three blank incubations (matrix without polymer) for each sampling interval;All incubation sets were stored in the dark at 25 °C.

CO_2_ levels in flasks from incubation set 1 were measured every 7 days using a universal gas analyser, the percentage of mineralisation pertaining to the carbon content of the initial sample was calculated, which provided data for the assessment of biodegradation of studied polymers.

Polymer specimens were collected after 1, 6 and 12 months for microscopic observation from incubation set 2 and three specimens were prepared for each sampling and polymer. Polymer samples were immediately observed by fluorescent microscopy and then stored in the fridge until analysis by scanning electron microscopy.

The mixture of polymer and soil together with the blank (1 g) was collected after 1, 6 and 12 months from incubation set 2 and DNA was isolated. DNA was also isolated from non-incubated soil (Day 0). DNA was then subjected to qPCR and the amount of biomass was calculated for each sampling. The same DNA was analyzed using next generation sequencing (NGS).

The fulvic and humic acids from set 3 were extracted after 3, 6 and 12 months by the modified NaOH method and their amounts were measured by spectrophotometer. The degree of their condensation was determined via spectrophotometry.

Additionally, the amount of carbon that was transferred from polymer to soil and biomass was calculated.

### 3.2. Polymer Materials

PE wax was obtained by CLARIANT in the form of small pellets (0.5–2 mm). PBAT was obtained by BASF, Germany, PHB Enmat Y 3000 by TianAn China and PLA 4042D by Nature Works. Polyester films were prepared by compression moulding (100 µm) and cut into 5 × 5 mm pieces.

### 3.3. Thermal Properties

The thermal properties of non-incubated samples were investigated by DSC in the Mettler Toledo DSC1 STAR system. The measurements were performed under a nitrogen atmosphere (50 cm^3^·min^−1^). The temperature ramp was set from 20 to 200 °C (10 K·min^−1^), followed by annealing at 200 °C for 5 min, subsequently followed by a cooling scan from 200 to 20 °C (20 K·min^−1^), then an isothermal step at 0 °C for 5 min, and finally a second heating scan from 0 to 200 °C (10 K·min^−1^). The melting point temperature (Tm) as well as the heat of fusion (ΔHm) were measured during the first heating cycle.

The degree of crystallinity χc was calculated according to the following equation (Equation (1):χ_c = 〖ΔH〗_m/(〖ΔH〗_m^0) × 100(1)
where 〖ΔH〗_m is the heat of fusion and 〖ΔH〗_m^0 is the tabulated heat of fusion for theoretically 100% crystalline Ecoflex homopolymer (115 J g^−1^) [53].

### 3.4. Gel Permeation Chromatography

Molecular weight and distributions were determined by GPC on the Breeze chromatographic system (Waters, Milford, MA, USA) equipped with a PLgel Mixed-D column (300 × 7.8 mm, 5 μm, Polymer Laboratories, Ltd., Church Stretton, UK) and the Waters 2487 dual-absorbance detector. Data were processed via Waters Breeze GPC software (Waters).

### 3.5. Soil Characterization

Soil LUFA 2.2 together with its characteristics was provided by LUFA Speyer, Germany. Soil contained 1.77 ± 0.56% organic carbon, 0.02 ± 0.06 nitrogen and the pH (0.01 M CaCl_2_) value was 5.6 ± 0.3.

### 3.6. Incubations

Incubations for polymer mineralisation analysis were prepared according to ASTM 5988 standard—CO_2_ evolution in soil but were miniaturized. The biodegradation tests were run in 500 mL binometric flasks with septa equipped with stoppers [54]. Polymer samples (50 mg), standard soil LUFA 2.2 (15 g dw), perlite (5.0 g) and mineral medium (7.5 mL) were mixed. Five replicates were run for each incubation with a polymer sample, along with five blank replicants. The blank was prepared by mixing standard soil LUFA 2.2 (15 g dw), perlite (5.0 g) and mineral medium (7.5 mL) and was incubated under the same conditions as incubations with the polymers. The sealed flasks were incubated at 25 °C in the dark.

Another set of incubations was prepared for analysis by microscopy and molecular biology. Biometric flasks (500 mL) were filled with 300 mg sample, 30 g standard LUFA 2.2 soil dw and 15 mL of mineral media. Incubations were prepared in duplicates together with the blank. The incubations were mixed every 7 days, which ensured enough oxygen. One gram of sample/wet soil mixture and also blank was sampled at several intervals during incubation (1, 6, 12 months) for DNA isolation. Samples, which were later observed by microscopy, were extracted from the soil after 1, 6 and 12 months and gently washed.

Separate incubations were also prepared for humic substance analysis, which were determined using the spectrophotometric method. The samples contained 30 g dw standard LUFA 2.2 soil, 150 mg sample, and 10 mL mineral media in each flask. The incubations were prepared in triplicate along with the blank. The incubations were mixed every 7 days, which ensured enough oxygen. The analysis of humic substances was performed after 3, 6 and 12 months.

### 3.7. Mineralisation

Headspace gas was sampled at appropriate intervals through the septum with a gas-tight needle and conducted through a capillary into the gas analyzer (UGA, Stanford Research Systems, Sunnyvale, CA, USA).

The concentration of CO_2_ was derived from the calibration curve, which was obtained by analysis of calibration gas mixture with declared composition. The percentage of mineralisation pertaining to the carbon content of the initial sample was calculated from the CO_2_ concentration found. The endogenous production of CO_2_ in blank incubations was subtracted to obtain values representing net sample mineralisation.

From the concentration determined, the percentage of mineralisation pertaining to the initial carbon content of the sample was calculated according to the equation:M = m_C/(m_(s)*w_C)
where M (%) is the percentage of mineralisation, mC (mg) is the mass of carbon evolved as CO_2_, ms (mg) is the weight of the polymer sample, and wC is the percentage (*w*/*w*) of carbon in the material investigated. The value of wC for each polymer was determined on a Flash elemental analyzer 1112 (Thermo Scientific, Waltham, MA, USA). The incubations were aerated monthly.

The polymer specimens were collected at selected time intervals (1, 6, 12 months) and immediately observed using fluorescent microscopy. The specimens, which were observed by SEM were stored in the fridge in sterile Eppendorf tubes.

Scanning electron microscopy

The materials were analysed using the Phenom Pro (ThermoFisher Scientific, Waltham, MA, USA) SEM. The samples were coated with gold/platinum alloy and observed at the acceleration voltage of 10 kV in the backscattered electron mode.

Fluorescent microscopy

Polymer samples were stained using a LIVE/DEAD^®^ BacLight Bacterial Viability Kit ((ThermoFisher Scientific) according to manufacturer instructions. Two stains, SYTO9 and propidium iodide were mixed with sterile distilled water. Polymer films were stained in this mixture for 15 min. Live and dead microorganisms were observed using an Olympus BX53M Upright Microscope (Olympus) in fluorescence mode.

### 3.8. Molecular Biology Methods

DNA was isolated using a DNeasy PowerSoil Kit (Qiagen, Hilden, Germany) from 1 g of fresh soil/sample mixture at selected time intervals (1, 3, 6 and 12 months) together with the blank and with fresh soil before incubation (Day 0). qPCR was used for the quantification of microbial biomass during incubation. Primers specific for bacteria (16S rRNA gene)—341F, 518R [55] and fungi—FR1, FF390 (18s rRNA gene) [56] were chosen to amplify the appropriate part of the genes, and cycle threshold (Ct) values were recorded.

DNA isolated from *Bacillus subtillis* (bacteria) and *Aspergillus niger* (fungi) served to create calibration curves for biomass quantification.

qPCR was performed by using thermocycler CFX 96 Real-Time (Bio-Rad, Hercules, CA, USA) with Luna Universal qPCR Master Mix (New England Biolabs, Ipswich, Mass, USA). The total reaction volume, 25 μL, included 12.5 μL of Luna mix, 250 nmol·L^−1^ of a forward primer, 250 nmol·L^−1^ of a reverse primer and 1–2 μg of DNA template.

The qPCR conditions were as follows: initial denaturing at 95 °C for 3 min, followed by 45 cycles each comprising 95 °C for 30 s of denaturing, 60 °C for 30 s of annealing, and 72 °C for 1 min of extension; the final extension was performed at 72 °C for 5 min. Data were normalized to GAPDH expression. Reference control and non-template negative controls (using water instead of DNA) were included in every run for both genes.

The baseline and cycle threshold were automatically calculated using the C1000 Touch Thermal Cycler equipped with a CFX 96 Touch™ System Software, version 2.1 (Bio-Rad, Hercules, CA, USA). The melt curve analysis was performed on the same device (CFX 96 Real-Time) after the completion of qPCR. Obtained PCR products of the MCO and GAPDH had melting temperatures of 76 ± 0.5 °C and 77 ± 0.5 °C, respectively.

The percentage of carbon transferred to biomass from the polymer samples was calculated for 12 months of data.

Previously isolated DNA was used to amplify specific regions of bacteria V3–V5 (16S) and fungi ITS2 (18S) rRNA genes using primers F357 (5′-CCTACGGGAGGCAGCAG-3′) and R926 (5′-CCGYCAATTYMTTTRAGTTT-3′), and ITS3F (5′-GCATCGATGAAGAACGCAGC-3′) and ITS4R (5′-TCCTCCGCTTATTGATATGC-3′), respectively, with barcodes and the universal overhang. Illumina sequencing adaptors were introduced in the second PCR, all in accordance with the Illumina instructions (Illumina. 16S Metagenomic Sequencing Library. Illumina.com 2013, No. B, 1–28.). Products were evaluated by agarose electrophoresis, quantified with a fluorimetric high-sensitivity Acugreen kit (Bioline) and pooled into a library. The sequencing library was sequenced on MiSeq (Illumina) using v2 version of chemistry and 250 nt paired end reads settings in the external laboratory (SEQme s.r.o., Dobříš, Czech Republic). The data were further processed with the DADA2 R package [57] and further visualized with the phyloseq R [58] and ComplexHeatmap R packages. Taxonomy was assigned for bacteria using the SILVA 132 SSU NR 99 reference database [59] and the 8.3 release of the UNITE reference database for fungi [60].

### 3.9. Humic Substances Analysis

To optimize and determine the reliability of the method humic acid (HA) reference material (humic acid technical standard material) was obtained from Sigma-Aldrich (Germany) and fulvic acid (FA) reference material (Suwannee River Fulvic Acid Standard II) was obtained from the International Humic Substances Society (IHSS, Denver, CO, USA). Elemental composition (C, H, N, O) of both HA and FA standards was determined on a Flash elemental analyzer 1112 (Thermo Scientific, Waltham, MA, USA).

The fulvic and humic acids from incubations with polymers and the blank were extracted using the modified NaOH method [61]. The extraction process of analysed HAs and FAs is shown in the scheme (Figure 13). Firstly, the incubated mixture of polymer and soil was leached using 0.1 M sodium hydroxide, using a flask flushed with N_2_ and closed with parafilm, in a ratio of liquid to solid (L/S = 10) for a minimum of 4 h shaking. The total volume of the solution was 300 mL. After that, the suspension was allowed to settle overnight, and the supernatant was collected using centrifugation at 3000× *g* for 10 min. Then, the total content of humic substances was measured. The obtained supernatant was then acidified to pH = 2 using a solution of 6 M hydrochloric acid with constant stirring. The suspension was allowed to stand for an extra 12 or 16 h and was later centrifuged (3000× *g*, 10 min) to separate the humic and fulvic acid contents. The supernatant was decanted, FA content was measured in fulvic acid solution (FA), and the precipitated HAs were redissolved in 0.1 M NaOH and analysed as HA content.

Humic substances were analysed using a spectrophotometer UNICAM UV500 (Thermo Spectronic, Cambridge, UK). The absorbances were measured in triplicate at multiple wavelengths of 350, 370, 400, 450 and 500 nm to determine the FA concentrations. The final concentration of the samples was calculated by averaging the measured values. The absorbances of HA content were measured at two wavelengths, 465 and 665 nm, and the final concentration of the samples was calculated in a similar manner. Using this method, the ratio between absorbance values at wavelengths 465 nm and 665 nm, which inversely characterise the degree of condensation of humic substances, was also determined [62].

## Figures and Tables

**Figure 1 ijms-23-15976-f001:**
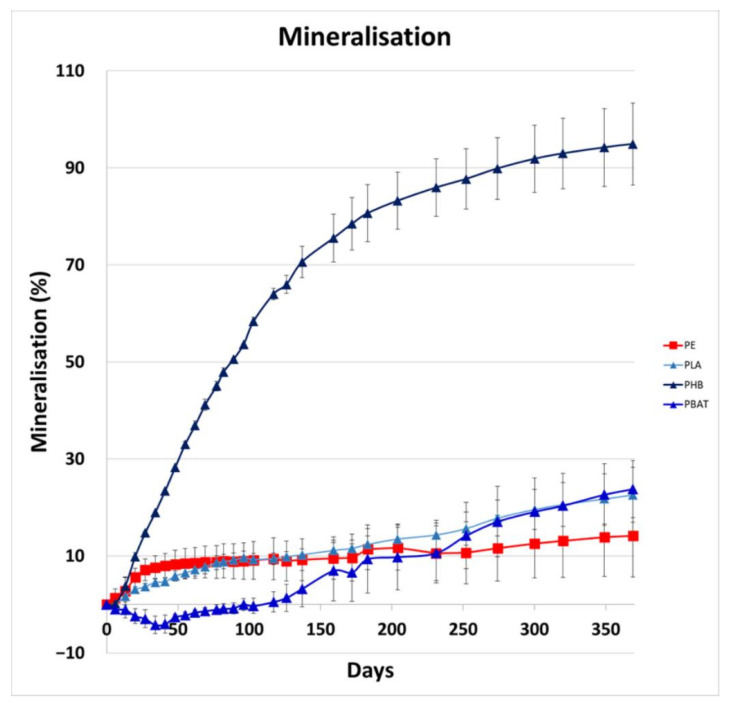
Mineralisation rate of tested samples.

**Figure 2 ijms-23-15976-f002:**
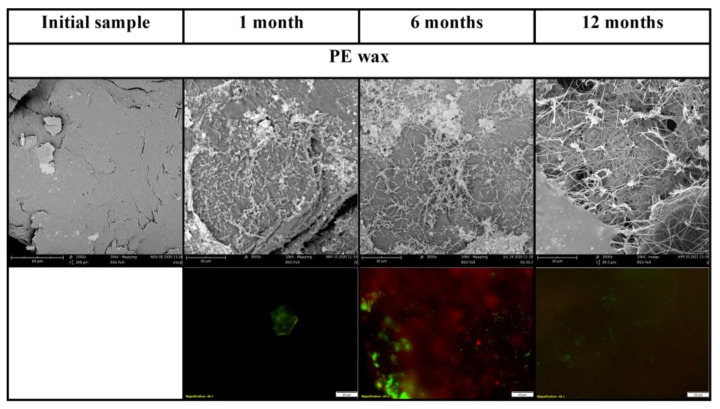
Scanning electron micrographs and live/dead photos of PE sample. Magnification of initial SEM samples 1000×, incubated SEM samples 3000×, fluorescent microscopy samples 400×.

**Figure 3 ijms-23-15976-f003:**
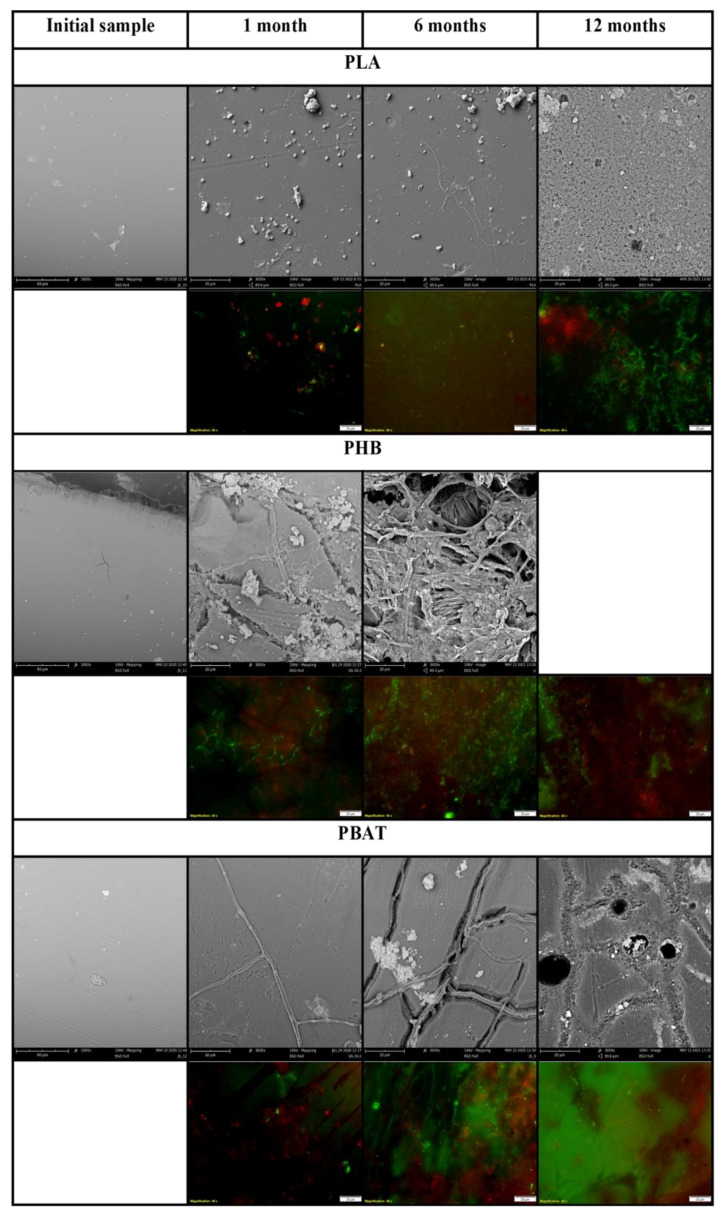
Scanning electron micrographs and live/dead photos of polyesters. Magnification of initial SEM samples 1000×, incubated SEM samples 3000×, fluorescent microscopy samples 400×.

**Figure 4 ijms-23-15976-f004:**
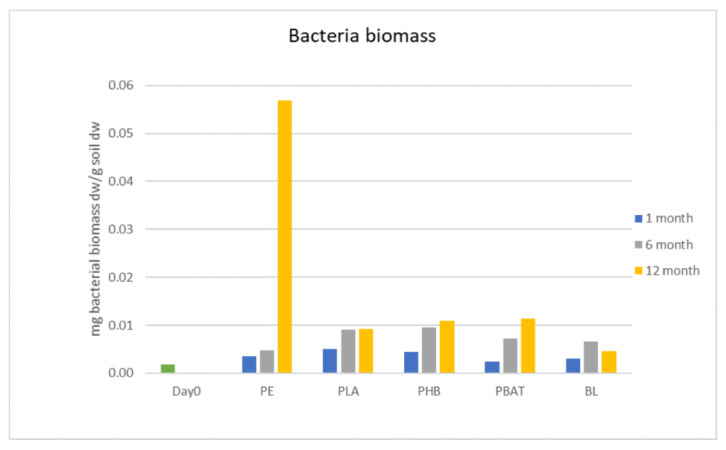
Biomass of bacteria in incubations.

**Figure 5 ijms-23-15976-f005:**
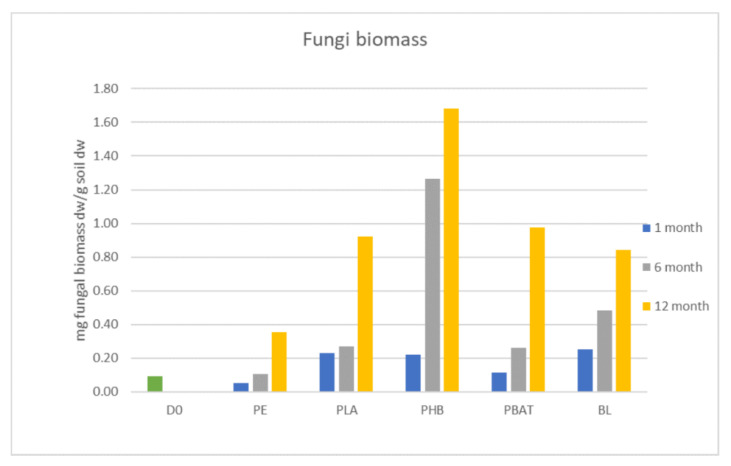
Biomass of fungi in incubations.

**Figure 6 ijms-23-15976-f006:**
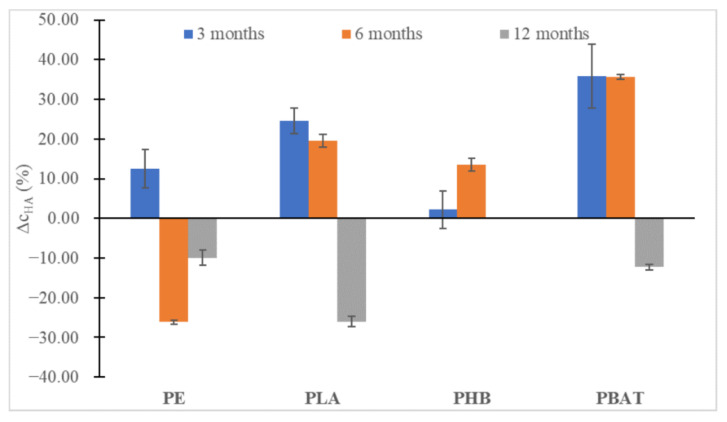
The percentual change in HA concentrations in the soil after 3, 6 and 12 months of incubation relative to HA content in blank incubation.

**Figure 7 ijms-23-15976-f007:**
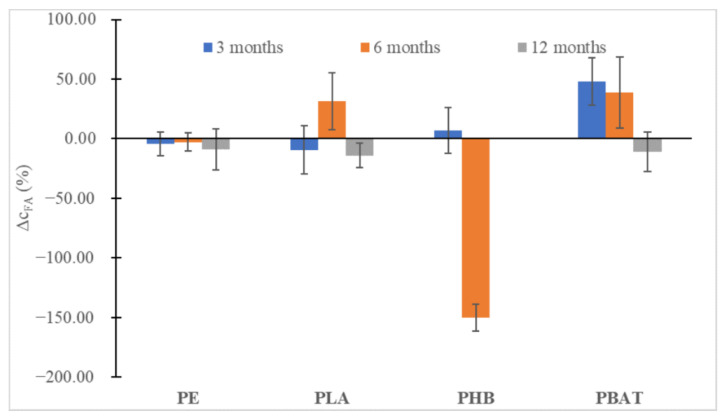
The percentual change in FA concentrations in the soil after 3, 6 and 12 months of incubation relative to FA content in blank incubation.

**Figure 8 ijms-23-15976-f008:**
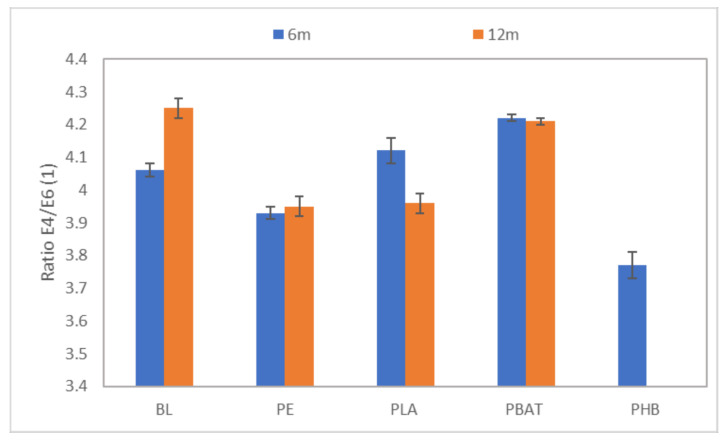
The ratio of spectral absorbances at 445 and 665 nm.

**Figure 9 ijms-23-15976-f009:**
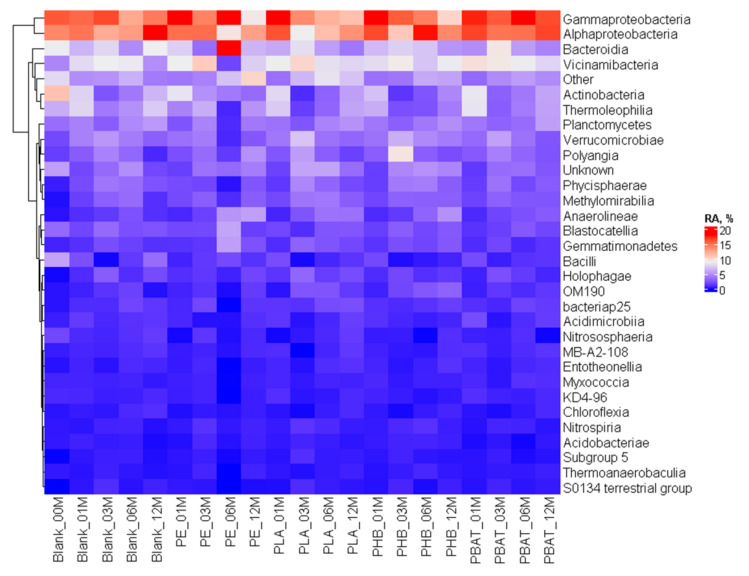
Heatmap of the bacterial community composition and time evolution at the class level.

**Figure 10 ijms-23-15976-f010:**
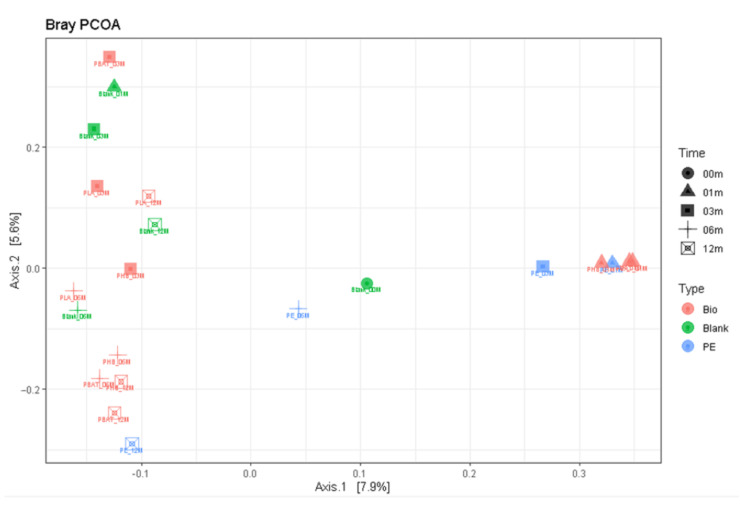
Principal component analysis scatter plot based on 16S rDNA of bacteria.

**Figure 11 ijms-23-15976-f011:**
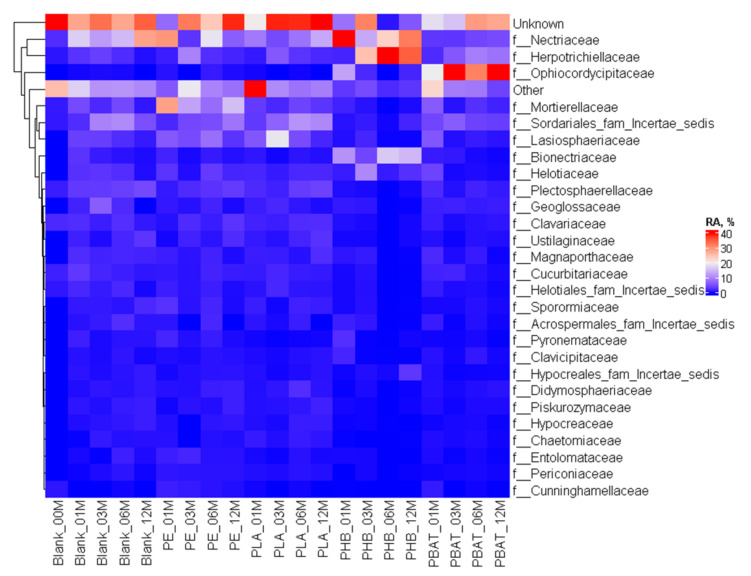
Heatmap of the fungal community composition and time evolution at the family level.

**Figure 12 ijms-23-15976-f012:**
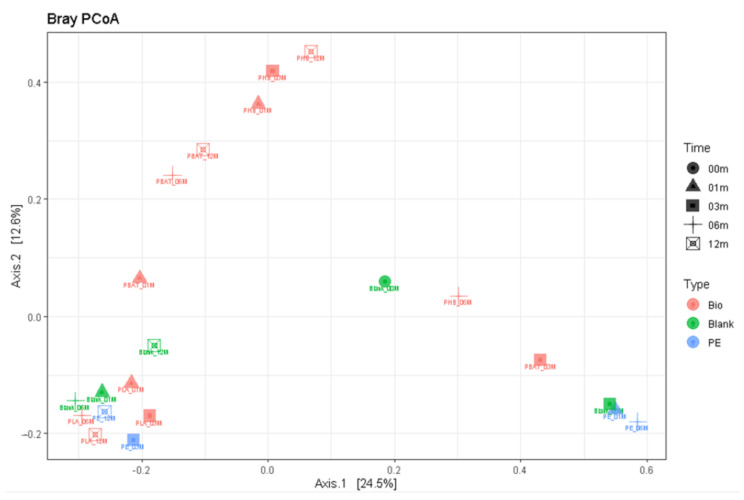
Principal component analysis based on 18S rDNA.

**Figure 13 ijms-23-15976-f013:**
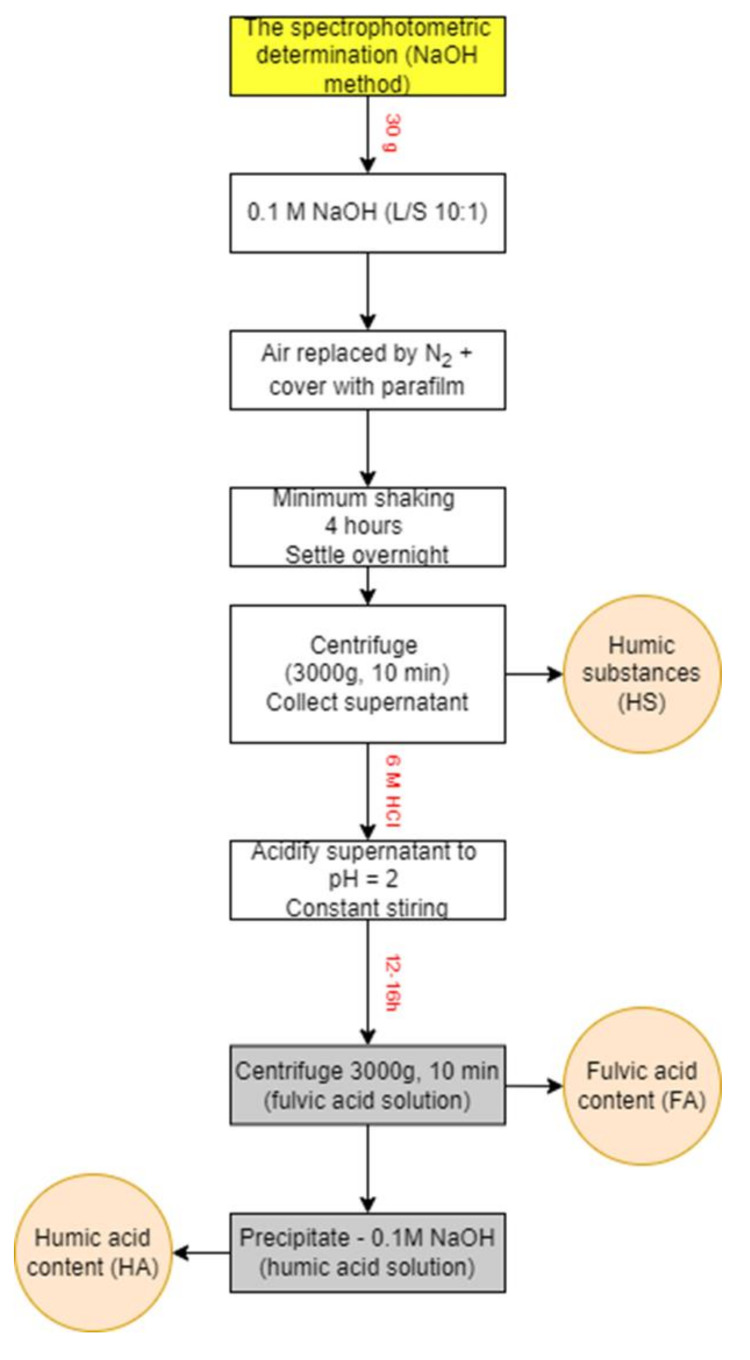
The scheme of spectrophotometric method (NaOH method).

**Table 1 ijms-23-15976-t001:** Properties of the samples.

Sample	Mw (g/mol)	Mn (g/mol)	Degree of Crystallinity (%)
PE	7900	2200	56.8
PLA	-	66,000	1.3
PHB	-	-	32.6
PBAT	93,000	35,700	9.9

**Table 2 ijms-23-15976-t002:** The elemental composition of commercial standards.

Sample	C (%)	H (%)	N (%)	S (%)
**Fulvic acid (FA)**	49.42 ± 0.1	4.41 ± 0.01	0.63 ± 0.01	0.31 ± 0.01
**Humic acid (HA)**	42.02 ± 0.2	3.46 ± 0.01	0.17 ± 0.01	0.90 ± 0.01

## Data Availability

Not Applicable.

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
