# Peer review of "Biodegradable Polyesters and Low Molecular Weight Polyethylene in Soil: Interrelations of Material Properties, Soil Organic Matter Substances, and Microbial Community"

_ijms, 2022, doi:10.3390/ijms232415976_

Round 1

Reviewer 1 Report

This is a pertinent study; the authors studied microplastic transformation and the interactions between non-biodegradable and biodegradable materials and soil organic matter and microbial community. The justification of the study is sound. Although standard methods were used in the analyses, the materials and methods section is not well-presented. Some key information is missing. The section should begin with experimental design so that a reader can see what will be done and in which order. In Line 329, “Five parallel flasks were run for each sample…” Please describe the “sample”. How many replicates were used for each sample? It seems that incubations and samples are used interchangeably. In line 336, “Samples, which were later observed by microscopy and DSC” How many samples. Please describe the “blank”. I do not see any statistical analysis. I understand one may not need statistical analysis for the SEM results, but the results for biomass and elemental composition, etc. need statistical analysis to make meaningful comparisons. The materials and methods should be improved before the manuscript is accepted.   

Other comments

Line 25: bacteria

Line 378: Please present scientific names throughout the manuscript in italics…Bacillus subtillis (bacteria) and Aspergillus niger (fungi) served

Reviewer 2 Report

The authors explain that biofilm formation on microplastic surfaces is a key point for minaralization.

Differences in materials (or substrate supply characteristics due to initial decomposition) are thought to affect the growth and adhesion of bacteria. UV weathering of plastics also affects miniaturization and adhesion of bacteria, so experiments under conditions close to the actual situation will be necessary.

--

2.2 Mineralisation of materials 

Line 95-96. Fig.1

Percentage of mineralization of the PBAT was negative during first 100 days. Author should make more detailed discussion regarding this phenomenon. Addition of PBAT might inhibit activity of soil bacteria ?

2.3 Microscopic observation

Line 134-138

The author may have performed fluorescence observation first, and then observed the same sample with SEM, but the method and steps of sample pretreatment should be described in Methodology.

Why was LIVE/DEAD staining performed instead of DAPI staining?

Filamentous bacteria or fungi growth was observed on the PE surface, but was not detected by LIVE/DEAD staining. Please explain whether this is because LIVE/DEAD (SYTO9, PI) staining is not effective for fungi. Also, line 136-138 states that most of the cells observed in LIVE/DEAD were green, but this is not the case with PE and should be corrected.

2.4. Biomass of microorganisms 

It is presumed that the activity of bacteria and fungi and the amount of biomass are closely related to the decomposition of the specimen during the experiment, that is, the amount of CO2 generated (Fig.1). Consideration should be given to this point.
